# Emergence of Nanoscale Drug Carriers through Supramolecular Self-Assembly of RNA with Calixarene

**DOI:** 10.3390/ijms24097911

**Published:** 2023-04-26

**Authors:** Ruslan Kashapov, Yuliya Razuvayeva, Nadezda Kashapova, Albina Ziganshina, Vadim Salnikov, Anastasiia Sapunova, Alexandra Voloshina, Lucia Zakharova

**Affiliations:** 1Arbuzov Institute of Organic and Physical Chemistry, FRC Kazan Scientific Center of RAS, 8 Arbuzov Str., 420088 Kazan, Russia; 2Kazan Institute of Biochemistry and Biophysics, FRC Kazan Scientific Center of RAS, 2/31 Lobachevsky Str., 420111 Kazan, Russia; 3Institute of Fundamental Medicine and Biology, Kazan (Volga Region) Federal University, 18, Kremlyovskaya Str., 420008 Kazan, Russia

**Keywords:** resorcinarene, RNA, supramolecular self-assembly, drug delivery, M HeLa, Chang liver

## Abstract

Supramolecular self-assembly is a powerful tool for the development of polymolecular assemblies that can form the basis of useful nanomaterials. Given the increasing popularity of RNA therapy, the extension of this concept of self-assembly to RNA is limited. Herein, a simple method for the creation of nanosized particles through the supramolecular self-assembly of RNA with a three-dimensional macrocycle from the calixarene family was reported for the first time. This self-assembly into nanoparticles was realized using cooperative supramolecular interactions under mild conditions. The obtained nanoparticles are able to bind various hydrophobic (quercetin, oleic acid) and hydrophilic (doxorubicin) drugs, as a result of which their cytotoxic properties are enhanced. This work demonstrates that intermolecular interactions between flexible RNA and rigid calixarene is a promising route to bottom-up assembly of novel supramolecular soft matter, expanding the design possibilities of nanoscale drug carriers.

## 1. Introduction

Despite the intensive development of anticancer therapy, cancer is one of the leading causes of death. The main reason for failure of therapy is a large number of genetic changes in cancer cells and their rapid growth, which leads to the emergence of drug resistance [1,2]. Due to the link between genetics and tumor formation, one of the promising strategies of the last decade is gene therapy and especially RNA therapy [1,3]. The RNA interference method involves the delivery of siRNA molecules to cells, which will suppress the expression of genes responsible for uncontrolled cell division [4], their viability [5], or resistance to chemotherapy drugs [6,7]. In addition, delivery of RNA into the cell can restore the function of tumor suppressor genes [8]. Of particular interest is the idea of creating RNA vaccines aimed at creating “antitumor immunity” [9,10]. However, the use of free RNA has a number of problems due to its specific behavior in the human body. RNA molecules degrade rather quickly in the body [11], and mRNA decay can help viruses avoid detection by the immune system [12]. At the same time, viral RNA can induce a strong immune response by activating interferons [13]. In addition, hydrophilicity and negative charge cause the inability of RNA molecules to penetrate biological membranes [14,15]. All this indicates the need to use special carriers (vectors) for the delivery of nucleic acids, and the nanoparticles of various nature can serve as such vectors. These can be both solid inorganic (quantum dots [16,17] or silicon nanoparticles [18]) and soft organic nanoparticles (liposomes [19,20,21], polymer nanoparticles [22,23,24]). Complexation of RNA with such carriers occurs predominantly via electrostatic interactions [25,26]. The decrease and recharging of negative RNA ions, as well as the presence of hydrophobic fragments in the composition of nanocarriers, contribute to better permeability of the complex through the cell membrane [15,27]. Moreover, nanoparticles containing RNA can also bind anticancer drugs, which increases the effectiveness of therapy by several times [28,29,30]. Thus, the hydrophobic core of micelles or the phospholipid bilayer of liposomes are able to solubilize hydrophobic chemotherapy drugs [6,31,32,33,34,35,36], which improves their aqueous solubility and thereby increases bioavailability. For doxorubicin hydrochloride, the formed complex is then solubilized into polymeric micelles based on clinically approved poly(ethylene glycol)-block-poly(D,l-lactide) [22].

One of the most widely studied classes of substances for gene transfection are surfactants. A special type of surfactant is macrocyclic amphiphiles based on three-dimensional cyclodextrins and calixarenes. Such macrocycles can have a hydrophilic head fragment formed by ionic groups located on one of the edges of the macrocyclic skeleton and a hydrophobic part formed by n-alkyl substituents on the opposite rim, which is a prerequisite for their aggregation into various nanostructures. Due to their low toxicity and lower aggregation threshold compared to conventional non-macrocyclic surfactants, macrocyclic amphiphiles are promising components for transfection [37]. In [38], porous nanospheres were obtained using RNA containing the EpCAM aptamer for targeted delivery and amino-beta-cyclodextrin, the hydrophobic cavity of which included sorafenib. After intravenous injection of this composition into mice with tumors, internalization was observed in target cells of hepatocellular carcinoma, where they were cleaved by cytoplasmic Dicer enzymes, releasing siRNA and sorafenib for synergistic therapy. By varying the mass ratios of the cationic dodecyl amino-beta-cyclodextrin with the anionic undecyl sulfo-beta-cyclodextrin, the size and charge of nanoparticles containing siRNA can be controlled. The presence of an anionic macrocycle contributed not only to a decrease in the zeta potential, leading to a decrease in toxicity, but to a 3-fold decrease in polydispersity, which contributed to better permeability through cell membranes [39]. In another study, dodecyl amino-beta-cyclodextrin was also used for the co-delivery of docetaxel and siRNA, which included folate lipid-based vesicular nanoparticles [40]. The obtained nanoparticles enhanced the apoptotic effect of the encapsulated drug, which significantly slowed down the growth of colorectal cancer in mice without causing significant toxicity in healthy mice.

In addition to cyclodextrins, calix-like macrocycles can also be used to bind RNA. Undoubtedly, the cationic nature of these macrocycles also plays a key role in binding to nucleic acids, and as a result of this electrostatic interaction, the stability of RNA against thermal denaturation is increased [41]. The polycationic argininocalix[4]arene is able to effectively deliver miRNA, pre-miRNA and anti-miRNA molecules to target cells, while maintaining their biological activity, and the delivery of pre-miRNA and anti-miRNA molecules is higher compared to reference gold standards. In addition, this study showed no effects of cytotoxicity and antiproliferative activity [42]. In another work, it was demonstrated that due to cooperative supramolecular interactions between yeast RNA and calixpyridinium, the spontaneous formation of stable and stimulus-sensitive aggregates is realized. RNase A induces the breakdown of these aggregates, which is accompanied by the release of pre-loaded drugs [43].

Most of the studies devoted to the binding of nucleic acids using macrocycles are focused on both non-covalent complexes with DNA and covalent conjugates with deoxyribonucleotides. In the work [44], calix[4]arene–nucleoside conjugates were synthesized, which aggregate into structures similar to natural oligodeoxyribonucleotides. The synthesized conjugates can be the building blocks of programmed oligonucleotide nanostructures and effective turning points in the long sequences of these nanostructures. There are very few works devoted to RNA binding, which prompted us to study the ability of calix[4]resorcinol-based cavitands to form joint aggregates with RNA. In this work, dodecyl calix[4]resorcinol, modified along the upper rim with viologen fragments (Figure 1), was chosen as the amphiphilic macrocyclic component. In our previous works, the low toxicity of this macrocycle [45] and its ability to form joint aggregates with anionic surfactants [46] were shown. These aggregates are able to act as carriers for doxorubicin and favorably change its antitumor activity. The choice of viologen-modified macrocycle is conditioned by the targeting ability of the viologen to mitochondria [47]. The metabolism of cancer cells often shifts from oxidative phosphorylation to aerobic glycolysis as the main generator of cellular ATP, which leads to an increase in the amount of NADH that can be oxidized by viologen groups [48]. The reduction of a dicationic viologen to a cationic radical and neutral forms may be accompanied by the release of a drug encapsulated in aggregates that were formed due to electrostatic interactions of viologen groups of macrocycles with oppositely charged molecules. As a model nucleic acid, a sodium salt of yeast RNA was taken, which is a specific stimulator of protein biosynthesis in hematopoietic and immunocompetent organs [49]. Using a set of physicochemical methods, the self-assembly in a mixed VR–RNA system was studied, the sizes of the formed particles were determined, and the effect of the ratio of components was evaluated. The resulting mixed aggregates were investigated as nanocarriers for biologically active substances of various solubility, namely, hydrophobic quercetin and oleic acid, as well as the hydrophilic antitumor drug doxorubicin hydrochloride.

## 2. Results and Discussion

### 2.1. Mixed Self-Assembly of VR and RNA

The first step in the supramolecular design of non-covalent conjugates based on VR and RNA began with the study of their co-aggregation using turbidimetry by measuring the optical density at 450 nm. Due to the rather high absorption of both pure components in the region of 220–300 nm, it was impossible to follow the changes in their characteristic absorption bands in the mixed system; therefore, the turbidimetric titration method was used to monitor the formation of joint VR–RNA aggregates in solution. There is no turbidity in pure RNA solutions (Figure 2a), but the presence of VR in an amount of 0.02 mg/mL in aqueous solutions of this biopolymer leads to the fact that the turbidity in these mixed solutions increases sharply with increasing RNA concentration until the equimass ratio VR:RNA = 1:1 is reached. The turbidity in a mixture with 0.02 mg/mL VR increases more slowly with a further increase in the RNA concentration and reaches a plateau at a 10-fold weight excess of RNA. In the opposite case, the addition of a variable amount of the macrocycle to a fixed amount of RNA (0.1 mg/mL) also shows that the mixed aggregates are not formed with a lack of VR (VR:RNA < 1:10) (Figure 2b). Further, the turbidity increases in the range of ratios 1:1 < VR:RNA < 1:10, and the light transmission does not change and reaches a plateau at a mass ratio of VR:RNA = 1:1. After reaching the maximum turbidity observed in the equimass solution, the light transmission in the solution does not decrease, but remains constant. This suggests that the joint aggregates formed in a mixed solution at a ratio of VR:RNA = 1:1 are stable and do not collapse with a further change in the ratio of components. Thus, the formation of joint VR–RNA aggregates occurs in an aqueous medium at ratios of VR:RNA in the range from 1:1 to 1:10.

The dynamic light scattering (DLS) method was used to determine the sizes of the formed joint aggregates. This method showed that the particle size in pure RNA solutions in the concentration range of 0.005–2 mg/mL does not exceed 8 nm (Figure 3a). The presence of a constant fraction of VR equal to 0.02 mg/mL in a mixture with RNA leads to the formation of aggregates 50–70 nm in size in mixed solutions even with a low biopolymer concentration, below 0.5 mg/mL (Figure 3b). Pure VR does not aggregate at a given concentration; however, its complex formation with RNA can reduce the repulsion of viologen groups of the macrocycle and promote aggregation at sufficiently low concentrations, which is of great importance in the development of non-toxic drug delivery systems. At an RNA concentration above 0.5 mg/mL, the particle enlargement of up to 100–170 nm and the appearance of the second type of particles with a size of 400–700 nm are observed, and the latter may indicate a possible adhesion of nonionic aggregates due to the complete electrostatic compensation of the viologen groups of VR by RNA phosphate groups. After such compensation, the free alkyl groups of VR are likely to initiate subsequent aggregation into larger aggregates.

The effect of various concentrations of VR on RNA solutions with constant biopolymer concentrations of 0.1 and 0.5 mg/mL was studied further using DLS (Figure 4). The minimum macrocycle concentrations at which joint aggregation is observed are 0.005 and 0.01 mg/mL for solutions containing 0.1 and 0.5 mg/mL RNA, respectively. A further increase in the fraction of VR does not lead to a significant change in the hydrodynamic diameters of the aggregates, and the aggregates with a size of 50 to 100 nm are formed regardless of the ratio of components in the mixed solution. The exception is equimass solutions with a ratio of VR:RNA = 1:1, in which precipitation was observed, indicating the smallest proportion of ionic fragments in these solutions. Thus, the DLS method showed that in order to obtain joint nanoscale aggregates based on the VR–RNA mixture, it is sufficient to add VR at a mass concentration 50 times lower than the concentration of RNA. The range of VR:RNA ratios in solutions for which it is possible to obtain a good correlation function and a relatively low polydispersity index (0.3–0.4) is 1:1–1:10, correlating well with turbidimetry data.

The ordering of the RNA structure in the presence of VR was confirmed by TEM. The TEM image for pure RNA (Figure 5a) shows filamentous fragments of the biopolymer [50]. These filaments may be non-polar and therefore generally aggregate in an aqueous medium, forming bundles that can be attracted to each other due to the hydrophobic effect [51]. The presence of an equimass amount of VR in an aqueous medium of RNA leads to the formation of more monodisperse spherical aggregates of a larger size (Figure 5b). The increased sizes of mixed aggregates obtained using TEM compared with DLS are due to the drying of the sample, during which the concentration increases. Nevertheless, comparison of images of pure RNA and RNA–VR mixture confirms the formation of joint nanoparticles due to the structural ordering action of the macrocycle.

Since the binding of VR to RNA is mainly due to strong electrostatic interactions between the negatively charged phosphate groups of the biopolymer and the viologen fragments on the upper rim of the macrocycle, the changes in the specific conductivity of solutions and the zeta potential of particles were further studied. The specific conductivity in a solution of pure RNA increases with an increase in concentration with an inflection at 1.2 mg/mL (Figure 6a). The presence of an inflection in the concentration dependence of specific conductivity in aqueous RNA solutions is explained by the fact that the number of negative charges of RNA increases with increasing concentration, but their mobility decreases due to aggregation, which is realized due to hydrogen bonds and π–π stacking [52]. It should be allowed that a decrease in electroconductivity can be additionally contributed by the partial condensation of macroions with an increase in their concentration and pairing. The resulting inflection value, corresponding to the critical aggregation concentration of pure RNA, does not correlate with the literature data (71.6 μg/mL), observed using turbidimetry [43]; however, according to other literature data, RNA aggregation is possible in the concentration range from 170 to 800 μg/mL [53]. Probably, RNA self-assembly consists of several steps, one of the subsequent steps of which was observed by us using the conductometric method.

When VR is added, the specific conductivity of mixed solutions becomes lower, and the resulting dependence has a greater slope compared to the dependence for pure RNA (Figure 6b), which is due to a decrease in the number of ionic fragments due to the electrostatic compensation of oppositely charged groups and a decrease in the mobility of joint aggregates due to an increase in their size. This dependence also has an inflection at an RNA concentration of 0.16 mg/mL, which corresponds to the mass ratio of VR:RNA = 1:8. Thus, a 7.5-fold decrease in the critical aggregation concentration in the mixed VR–RNA system compared to pure RNA is probably due to the effect of eight positive charges on the upper rim of the macrocycle, which enhances aggregation with RNA.

Due to the presence of a large number of phosphate groups, an increase in the proportion of RNA in a mixture with a constant VR concentration of 0.02 mg/mL leads to a decrease in the zeta potential of the particles towards negative values (Figure 7a). In an equimass solution VR:RNA = 1:1 leads, the formation of particles with a positive zeta potential is observed and the particles with a negative zeta potential are formed at a 2.5-fold excess of RNA in the solution. A slightly different phenomenon was observed during an increase in the proportion of VR in the mixture with a constant RNA concentration of 0.1 mg/mL. The higher absolute value of the zeta potential in the mixture with an excess of RNA than in the mixture with an excess of VR is probably related with a higher constant concentration of one of the components. Despite this, an increase in the proportion of VR in a mixture with a constant RNA concentration of up to a ratio of VR:RNA = 1:1 leads to the formation of more electrically neutral particles than in the case of the formation of particles in a mixture with a constant VR concentration when the ratio VR:RNA = 1:1 is also reached. Such a different behavior in mixed solutions with an increase in the proportion of one of the components of the supramolecular system is related with the conformational mobility of the component, the proportion of which is constant in the mixture. A relatively rigid macrocyclic skeleton creates steric obstacles to the interaction of VR and RNA, which leads to a positive value of the zeta potential of particles formed in an equimass solution at a constant concentration of VR and a variable proportion of RNA. Unlike the VR, a single-stranded nucleic acid is more flexible, as a result of which an increase in VR concentration in a mixture with a constant RNA concentration leads to biopolymer neutralization at a ratio of VR:RNA = 1:1 (Figure 7b). In general, the area of recharge in both mixed VR–RNA systems corresponds to the area of maximum turbidity formation observed by turbidimetry (Figure 2).

Morphological changes in RNA molecules under the action of VR can be traced using circular dichroism spectra, which are recorded to confirm the change in polynucleotide conformation as a result of VR binding. The spectrum containing negative bands at 209 and 239 nm, as well as a positive maximum at 268 nm, indicates that the RNA in the pure solution is in the A-conformation (Figure 8a). The addition of the first portions of VR (up to 0.03 mg/mL) to a mixture with a constant RNA concentration of 0.1 mg/mL does not lead to significant changes in the CD spectrum, but the presence of 0.06 mg/mL of VR causes a decrease in dichroic absorption and a slight shift of the signal to the long wavelength side of the spectrum (Figure 8b). A decrease in intensity and a slight red shift of the peak at 268 nm in a mixture with VR indicates a conformational change from the A-form to the B-conformation [50], which indicates the expansion of the major groove in the regular A-helix of RNA upon binding to VR. At the ratio of VR:RNA = 1:1, ellipticity almost disappears due to the formation of a poorly soluble complex. An increase in the proportion of VR above 0.1 mg/mL in a mixture with a constant RNA concentration of 0.1 mg/mL led to a reverse increase in the dichroic absorption of the positive and negative bands. Such changes in the CD spectra, together with the shift of the bands, served as evidence of a change in the secondary structure of RNA due to interaction with VR. Since an increase in the content of VR to 0.03 mg/mL in mixed solutions with a constant RNA concentration of 0.1 mg/mL caused an unsystematic change in dichroic absorption, which, apparently, was due to a measurement error, further circular dichroism spectra were obtained in mixed solutions with a higher constant RNA concentration (0.5 mg/mL) until the equimass ratio is reached (Figure 8c). There is a clear trend towards a decrease in the absorption bands at 209 nm and 268 nm by the same value with an increase in the proportion of VR in these mixtures (Figure 8d), which probably indicates a change in the orientation of RNA under the action of VR [54]. Thus, the revealed changes in ellipticity can be related with the compaction of RNA helices due to the electrostatic interaction between positively charged VR molecules and negatively charged RNA fragments.

^1^H NMR spectroscopy has been used to confirm intermolecular interactions between VR and RNA in an aqueous medium. Since the equimass complex VR:RNA = 1:1 precipitated in aqueous solution due to the high concentrations required to obtain ^1^H NMR spectra, a mixture with a slight 1.5-fold excess of VR was examined using ^1^H NMR (Figure 9). When comparing the ^1^H NMR spectrum of this mixture (Figure 9b) with that of pure VR (Figure 9a), a decrease in signal intensity is observed as a result of co-aggregation. The largest chemical shift by 0.04 ppm is experienced by viologen protons distant from positively charged nitrogen atoms, which may be due to a change in the environment of viologen ions as a result of electrostatic interaction with RNA. The upfield shift of protons of the VR terminal methyl group by 0.03 ppm and proton broadening of VR dodecyl chain in the presence of RNA indicate the participation of macrocycle alkyl chains in interactions with RNA, leading to joint aggregation.

To further confirm the intermolecular interaction between RNA and VR using IR spectroscopy, their complex in the solid state was prepared and isolated (Figure 10). Based on the literature data [55,56], the absorption bands of RNA can be interpreted as follows: a wide absorption band in the range of 3200–3600 cm^−1^ includes an O–H (~3400 cm^−1^) and N–H bond (~3100 cm^−1^ and ~3300 cm^−1^); the band at 2949 cm^−1^ corresponds to the stretching vibrations of the C–H bond; bands in the range of 1400–1700 cm^−1^ cover various stretching and bending vibrations of the C=C, C–N, C=O bonds included in the structure of nitrogenous bases; the antisymmetric and symmetric PO_2_ stretching bands correspond to the bands at 1237 and 1072 cm^−1^; bands in the region of 800–900 cm^−1^ correspond to vibrations of the sugar–phosphate backbone. A pure VR is characterized by bands of bending vibrations of the C–H bond at 3400 cm^−1^ and stretching vibrations of the same bond at 3117, 3038, 2923 and 2853 cm^−1^ [45]. The bands at 1639 and 1561 cm^−1^ correspond to the stretching vibrations of the C–N and C–C bonds. The bending vibrations of the C–H bond correspond to the band at 1451 cm^−1^, and the bending vibrations of the N^+^–CH_3_ bond in the viologen fragments correspond to the band at 829 cm^−1^. The IR spectrum of the mixture shows band shifts corresponding to complex formation between RNA and VR. The changes observed in the frequency of vibrations of the PO_2_ band are related to the electrostatic interaction of VR with RNA phosphate groups. At the same time, there is a shift in the absorption bands of nitrogenous bases and a change in the shape and width of the band corresponding to N–H and O–H bonds, which may indicate the existence of hydrogen bonds between the RNA bases and the viologen macrocycle.

### 2.2. Binding of Biologically Active Substances in VR–RNA Aggregates

The VR–RNA aggregates were investigated as nanocarriers of biologically active substances of various solubility by studying the spectrophotometric solubilization of these substances, which were chosen as hydrophobic quercetin and oleic acid, as well as hydrophilic doxorubicin hydrochloride. Quercetin in pure water does not give any absorption in the 380 nm region due to its lack of dissolution (Appendix A). Pure RNA is able to bind a small amount of quercetin, and a sharp increase in the soluble flavonoid detected by an increase in the absorption of quercetin at 380 nm is observed after reaching a concentration of 1.0 mg/mL (Figure 11a and Appendix A), which is probably due to the formation of hydrophobic domains in the composition of RNA aggregates [57]. At about this concentration, the conductometric dependence shows an inflection caused by the formation of aggregates (Figure 6a), and this process of quercetin binding occurs by the intercalation of its molecules between RNA base pairs, which is reflected in the appearance of quercetin fluorescence with a bathochromic shift (Appendix A) [58]. The presence of 0.022 mg/mL VR in an aqueous medium with RNA leads to an almost twofold increase in the aqueous solubility of quercetin when compared with pure RNA (Figure 11a and Appendix A). However, the amount of quercetin soluble in the mixture with VR:RNA ratios up to 1:10 is almost identical to the solubilization capacity of pure VR. In the mixture with more than a tenfold excess of RNA, an increase in the solubilization of quercetin is caused by the influence of the free fraction of RNA that did not interact with VR. When comparing the solubilization capacity of pure VR and its mixture with a constant RNA concentration of 0.1 mg/mL, the same binding of quercetin was also observed (Figure 11b and Appendix A). At an equimass ratio of VR:RNA, a decrease in the solubilization of quercetin is observed compared to pure VR due to a precipitate, the formation of which can be caused by a greater proportion of the constant concentration of RNA. Accordingly, in mixed VR–RNA systems the quercetin solubilization is observed at the level of the solubilization capacity of a pure macrocycle, i.e., capture of quercetin into nanoparticles formed in an aqueous medium at ratios of VR:RNA in the range from 1:1 to 1:10 occurs due to its solubilization in the hydrophobic domain of VR. To further confirm the binding of quercetin in aggregates based on VR and RNA, 1H NMR spectroscopy was used (Figure 9c). When comparing the ^1^H NMR spectra of VR–RNA mixtures in the absence (Figure 9b) and in the presence of quercetin (Figure 9c), the largest chemical shift by 0.06 ppm is observed for protons of alkyl chains VR, underscoring the location of quercetin in the hydrophobic domains formed by these chains.

After checking that the lipophilic quercetin is solubilized by VR–RNA nanoparticles, it was further decided to try to use these nanoparticles to bind a more hydrophobic and long-chain oleic acid (OA) molecule. Of the three components (VR, RNA, OA), only the macrocycle has a specific absorption at a wavelength of about 260 nm, so this absorption was used to study the interaction of OA with mixed VR–RNA systems. A very interesting effect of the amount of added OA on the absorption of VR is found by varying the ratio of VR:RNA. In a mixed solution with an equimass ratio of VR:RNA, the presence of OA causes a significant hypochromic effect until an OA concentration of 0.006 mM is reached (Figure 12 and Appendix A). This amount of OA correlates with the molar concentration of VR, which suggests that the macrocyclic fragment of joint nanoparticles, as in the case of quercetin, is the site of OA binding. According to the comparison of the ^1^H NMR spectra of OA in the absence (Figure 9e) and in the presence of the VR–RNA mixture (Figure 9d), the largest change in the chemical shifts of alkyl chain protons among other VR protons in the presence of OA and the greater change of all OA protons in the presence of VR–RNA mixture strongly confirm the binding of OA in aggregates based on VR–RNA due to solubilization into hydrophobic domains. In the OA concentration range above 0.006 mM, the maximum absorption intensity does not change, and the dependence of the absorption intensity at 260 nm reaches a plateau. It is interesting that such spectral changes were not observed in the UV spectra of pure VR and pure RNA, as well as the mixture with a two- and ten-fold excess of RNA. The probable reason for the different binding ability of OA by VR–RNA mixtures is the value of the fraction of VR, which determines the zeta potential of mixed nanoparticles and the number of hydrophobic domains in the composition of these particles. A larger amount of VR in the mixture with RNA makes it possible to give a more positive zeta potential, which determines the electrostatic attraction with carboxyl group of OA. It is likely that VR changes the structure of RNA not only through electrostatic interaction due to viologen fragments, but also through a hydrophobic effect due to alkyl fragments that change the stacking of purine and pyrimidine bases [59]. Thus, VR and RNA molecules interact with each other in equivalent weight amounts with the formation of a single supramolecular system capable of binding OA to a greater extent than pure macrocycle and pure biopolymer. The interaction of VR with RNA is realized due not only to the electrostatic attraction of the viologen groups of VR to the phosphate groups of RNA, but also due to a possible hydrophobic effect, which causes an increase in the lipophilic part within joint aggregates capable of accommodating OA.

After ascertaining that VR–RNA nanoparticles are able to bind hydrophobic molecules, it was further decided to test the ability of these nanoparticles to interact with hydrophilic anticancer drug doxorubicin hydrochloride due to electrostatic interactions with RNA [22,60], which is identified as fluorescence quenching of the chemotherapy drug (Appendix A). Since VR and DOX are similarly charged, they will compete with each other for complex formation with RNA. To determine the composition of mixed VR–RNA systems suitable for DOX binding, fluorimetric titration of a solution of 0.01 mg/mL RNA and 0.02 mM DOX with a solution of VR was carried out (Figure 13). The addition of the first portions of the macrocycle until the ratio of VR:RNA of 1:10 does not lead to an increase in the intensity of DOX fluorescence, but an increase in the emission of DOX fluorescence is observed at a higher concentration of VR near the equimass ratio due to its release from the complex with RNA. A further increase in the concentration of VR does not lead to a change in the fluorescence intensity of DOX, which confirms the achievement of the complete release of DOX by adding a fraction of VR equimass to the content of RNA. Thus, the optimal DOX delivery system is the composition VR:RNA = 1:10, in which RNA is capable of simultaneously binding DOX and VR.

Binding of DOX was reiterated by comparing its ^1^H NMR spectrum (Appendix A) with that of its mixture with RNA (Appendix A). The intensity of most DOX proton signals in this mixture was reduced due to electrostatic interaction with RNA, and some DOX proton signals were broadened and overlapped by broad RNA proton signals. The only clearly observed high-field shift of the methyl group near the positively charged amino group of DOX indicates its binding to RNA. The presence of VR in a tenfold deficiency relative to RNA in the composition of DOX did not affect the value of this chemical shift, which confirms the bound state of the drug in aggregates based on VR–RNA and correlates with fluorometry data (Appendix A). Unfortunately, increasing the proportion of VR in this composition led to precipitation due to the high concentrations of the components required to obtain ^1^H NMR spectra.

A fluorimetric study of the release of DOX at 37 °C in phosphate (PBS, pH 7.4) and acetate (ABS, pH 5.0) buffers simulating physiological media of normal and tumor cells, respectively, was performed using dialysis for the selected composition 0.02 mg/mL VR–0.2 mg/mL RNA–0.02 mM DOX. The experiment showed that the release of DOX into acetate buffer occurs approximately two times faster than into phosphate buffer (Figure 14). In addition, the analysis of the results of this study indicates a prolonged release of DOX into the phosphate buffer medium. During the first five hours, 20% of the drug is released in an acidic environment, and only 8% in a neutral environment. The output of time dependences of the percentage of DOX release to a plateau was observed after 20 h, and 38% of the loaded drug was released in a neutral medium and 73% in an acid medium The accelerated release of DOX in an acidic environment is probably due to the protonation of the nitrogenous bases of RNA, electrostatic interactions of which with a negatively charged phosphate backbone can increase the stability of the tertiary structure of RNA [61]. According to the literature data [62], an acidic environment stabilizes key intramolecular bonds of RNA, which may well initiate the release of DOX due to the competitive electrostatic interaction of phosphate groups with amino groups of RNA.

### 2.3. Biological Properties

The study of biological properties included the determination of the cytotoxic effect of the compositions in relation to normal (Chang liver) and tumor (M HeLa) cells. For pure RNA and its mixtures with a constant VR concentration of 0.02 mg/mL, the IC_50_ values in both cell media exceed 0.01 mg/mL; these solutions show no cytotoxic effect, regardless of the amount of RNA (Table 1). Binding of hydrophobic quercetin and OA in mixed solutions with a tenfold excess of RNA did not lead to a change in cytotoxicity against both cell lines. Further, mixed solutions with DOX were studied in the absence and presence of antioxidants capable of reducing the chemotherapeutic cardiotoxicity of DOX, so the IC_50_ values for these solutions were calculated from the concentrations of both RNA and DOX (Table 1). Since the study of the binding ability of hydrophilic DOX revealed complete capture of DOX by the mixed composition with a tenfold excess of RNA (Figure 13), nanoparticles of this composition with DOX in the absence and presence of antioxidants were tested for cytotoxic activity, cellular uptake, and cellular location. As can be seen from Table 1, free DOX exhibits greater cytotoxicity against tumor cells than normal cells, but when combined with RNA alone, its antitumor activity is almost halved. The binding of DOX in VR–RNA nanoparticles enhances the cytotoxic effect on M HeLa and Chang liver. However, the presence of quercetin had an antagonistic effect on DOX in M HeLa cells, which is consistent with the literature data [63], while the presence of OA, on the contrary, increased the cytotoxic effect on M HeLa. The probable reason for such an enhanced cytotoxic effect is the effect of OA on the fluidity of cell membranes, which increases in the presence of unsaturated fatty acids in the composition of the membranes. The cell membrane becomes more permeable to water and other small hydrophilic molecules with an increase in fluidity, and the rate of lateral diffusion of integral proteins also increases.

The intracellular content of DOX that entered M HeLa tumor cells both in free form and in various formulations in the absence and presence of antioxidants was further studied. Appendix A shows the obtained data on the intensity of DOX fluorescence, which was used to judge the content of this drug in cells. First of all, the fact of less intense fluorescence of free DOX compared to its formulations attracts attention, despite the same amount of DOX in each sample. Since the fluorescence of DOX in the presence of both pure RNA (Appendix A) and mixture with a small amount of VR (Figure 13) is lower compared to free DOX, the probable cause of the increased intensity detected using a flow cytometer is the increased affinity of a number of proteins contained in the cytoplasm of M HeLa cells to RNA [64]. In addition, fluorescence microscopy with the DAPI dye, which specifically binds to the DNA of cell nuclei, revealed the presence of apoptotic cells under the action of three compositions of RNA–DOX, VR–RNA–DOX and VR–RNA–DOX–Q (Figure 15). When using these three samples, a sign of apoptosis is observed, expressed in the presence of brightly glowing condensed chromatin in the images. The inhibition of cell viability is due to the mechanism of action of DOX manifested itself in the ability to bind and inhibit the enzyme topoisomerase II, which disrupts DNA repair and leads to apoptosis. Fluorescence microscopy experiments allowed us to evaluate the development of apoptotic effects in M HeLa cells induced by DOX both in free and drug-bound forms, and the latter contributed to the intensification of apoptotic processes in these cells.

Interestingly, for the composition with OA, which exhibits the greatest cytotoxic effect against M HeLa (Table 1), weak penetration into these cells is observed (Figure 15). The probable reason for this phenomenon is the fixation of DOX in the composition of mixed aggregates due to additional CH–π interactions between the alkyl fragment of OA and the aromatic fragment of DOX [65]. Due to this additional supramolecular interaction, the presence of OA, which is capable of exerting a stabilizing effect [66], in the mixed VR–RNA–DOX system prevents the penetration of DOX into the cell nuclei due to the strong binding of the drug in the composition of nanoparticles. However, the increased cytotoxicity of this system clearly indicates the intracellular localization of nanoparticles, which can be localized to a greater extent in mitochondria and to a lesser extent in plasma membranes, microsomes and cytoplasm [67].

## 3. Materials and Methods

### 3.1. Chemicals and Solution Preparation

Details of the synthesis of VR were described by Kashapov et al. [45]. RNA sodium salt from yeast (Thermo Scientific), quercetin (95%, Acros Organics) and oleic acid (99%, Alfa Aesar) were used as received. Sample solutions were prepared in deionized water (18.2 MΩ) collected from a Millipore Direct-Q 5 UV water purification system. The pure biopolymer and macrocycle solutions were prepared by dissolving a solid sample of the substance in a certain volume of this water and then diluting the concentrated stock solution. The mixed RNA–VR solutions were prepared by simply mixing different volumes of aqueous stock solutions of RNA and VR in a certain ratio at room temperature.

### 3.2. Absorption Spectrophotometry

Optical density values measured during turbidimetric titration were obtained using a Specord 250 Plus spectrophotometer (Analytic Jena, Jena, Germany) for aqueous solutions of samples in the 1-cm quartz cuvettes. UV–vis absorption spectra were recorded using the same spectrophotometer. To determine the solubilization of quercetin, an excess amount of solid quercetin was added to aqueous solutions of pure RNA, VR and mixed RNA–VR samples placed in vials. Aqueous solutions of the samples were stored at room temperature of 25 °C for 2 days, during which the solutions were periodically mixed on an orbital shaker at a temperature of 25 °C and a stirring speed of 250 rpm for 3 h to prepare a saturated solution, and then absorbance of quercetin at 380 nm (molar extinction coefficient is 23,487 L·mol^−1^·cm^−1^) was measured. The absorption of quercetin was monitored during the next 5 days of storage at room temperature and no change in this absorption was found.

### 3.3. Fluorescence Spectroscopy

Fluorescence emission spectra of doxorubicin hydrochloride (DOX) were measured on a Hitachi F-7100 spectrophotometer (Hitachi, Japan) at 25 °C with an excitation wavelength of 481 nm. Both excitation and emission slit widths were kept at 10 nm.

### 3.4. Dynamic and Electrophoretic Light Scattering

The measurement of hydrodynamic diameter values was conducted using a Zetasizer Nano instrument (Malvern, UK) equipped with a 4 mW He-Ne laser operating at 632.8 nm. Correlation data were fitted using the cumulants method to the logarithm of correlation function, yielding the diffusion coefficient. The backscattered light was detected at 173°, and the number-average hydrodynamic diameter was calculated using the Stokes-Einstein equation. For the measurement of zeta-potential, the same instrument with laser Doppler velocimetry and phase analysis light scattering was used. All light scattering measurements were repeated at least five times at 25 ± 0.1 °C, and the data acquired were processed using the Malvern Zetasizer software.

### 3.5. Transmission Electron Microscopy

The morphology of pure RNA and mixture RNA–VR aggregates was examined using electron microscopy with a Hitachi HT7800 transmission electron microscope (Hitachi High-Tech Science Corporation, Tokyo, Japan). Sample for imaging was prepared in deionized water; 5 µL of the solution was applied straight onto a 3.05 mm in diameter copper grid with a formvar film (01700-F, Ted Pella, Inc., Redding, CA, USA) and dried at room temperature. The grid with the dried sample was placed in the transmission electron microscope using a special holder, followed by imaging at an accelerating voltage of 80 kV in the TEM mode.

### 3.6. Circular Dichroism (CD)

Far-UV CD spectra from 190 to 260 nm were recorded using a JASCO J-1500 spectropolarimeter (Jasco Corp., Tokyo, Japan) using a 1-mm quartz cuvette. The scan parameters were the following: 100 nm/min scan rate, 0.1 nm step size, 1 nm bandwidth and 4 s D.I.T. The CD spectra obtained are the average of three measurements minus the solvent baseline.

### 3.7. Conductometry

The specific conductivity values were obtained using an Inolab Cond 720 conductometer (WTW GmbH, Wuppertal, Germany) with a graphite electrode with a cell constant of 0.475 cm^−1^ ± 1.5%. These values were measured at least three times for each sample, resulting in the values varying within 2%. All samples were studied at 25 ± 0.1 °C.

### 3.8. FTIR-Spectroscopy

The IR absorption spectra were recorded in KBr pellets on a Vector-22 FTIR spectrometer (Bruker) with a resolution of 4 cm^−1^ and 16 scan accumulation. To prepare the solid-state RNA–VR complex, VR (1 mg) was dissolved in water (1 mL) and RNA (10 mg) was added. The solution was stirred at room temperature for 2 h, then the precipitate was separated by centrifugation, washed 3 times with water, and dried under vacuum to yield 10.4 mg (95%) of yellow solid.

### 3.9. NMR Spectroscopy

The ^1^H NMR spectra were recorded using a Bruker AVANCE(III)-600 spectrometer (Germany) operating at 600.1 MHz, which is equipped with a 5 mm broadband inverse probe head with z-gradient accessories to produce a field gradient up to 50 G·cm^−1^. Samples were prepared in D_2_O. The spectra were recorded at 303.0 ± 0.2 K. The chemical shifts are reported in the ppm scale and referred to the solvent (δ(HDO) 4.7 ppm).

### 3.10. Substrate Release

To quantify the release profile of DOX from VR–RNA aggregates, the dialysis using a Slider-A-Lyzer membrane (3 mL, 2 kDa, Thermo FS) was carried out. DOX release was studied in acetate (pH 5.05) and phosphate (pH 7.4) buffers at 37 °C. Phosphate buffer was prepared using PBS tablets (Sigma-Aldrich: St. Louis, MO, USA). To prepare the sodium acetate buffer, 2.886 g of anhydrous sodium acetate was dissolved in 400 mL of deionized water, 0.848 mL of glacial acetic acid (ρ = 1.0492 g/cm^3^, i.e., 0.889 g) was added, adjusted to pH 5.0 with HCl, then diluted with water to 500 mL. The volume of the sample containing DOX was 4 mL, the volume of the external buffer solution was 250 mL. The release of DOX was monitored fluorometrically by its content in the external buffer. The DOX concentration was determined from the calibration dependences of the fluorescence intensity at 555 nm (Appendix A).

### 3.11. Cytotoxic Activity

Aqueous solutions of various compositions were tested for cytotoxicity in human normal (Chang liver) and tumor (M HeLa) cell lines. The cytotoxicity of the nanoparticles on human cancer cells was assessed and was determined by means of the MTT test. The degree of cell growth inhibition was calculated, and then the concentration causing 50% cell growth inhibition (IC_50_) was determined from the curve of cell culture growth versus concentration. The IC_50_ values were calculated using the online calculator MLA—Quest Graph™ IC50 Calculator (AAT Bioquest, Inc., Sunnyvale, CA, USA) (accessed on 1 February 2023). The experiments were repeated three times, and the results are presented as mean ± standard deviation.

### 3.12. Flow Cytometry

Flow cytometry was used to assess drug uptake by cells. Untreated cells were used as a negative control. Cells were seeded in 24-well plates at 5 × 10^4^ cells per well and incubated for 24 h. The cells were then cultured with the solutions to be tested. Fresh culture medium without test compositions was added to control cells. After treatment, the cells were washed three times with phosphate buffered saline to remove any free compositions. The cells were then trypsinized and resuspended in fresh phosphate buffer with 10% fetal bovine serum. Cell suspensions were analyzed using a Guava easy Cyte (Merck Millipore, Burlington, MA, USA).

### 3.13. Fluorescence Microscopy

Cell nuclei were stained with DAPI. After treatment with the studied compositions, photography was carried out using a Nikon Eclipse Ci-S upright motorized fluorescence microscope (Nikon, Nanjing, China) at ×400 magnification.

## 4. Conclusions

In this work, supramolecular interactions of RNA with the amphiphilic viologen calix[4]resorcinol were studied. Due to the cooperative effect of electrostatic and hydrophobic interactions between them, the spontaneous formation of nanoparticles in an aqueous medium at room temperature and without the use of ultrasonic treatment is realized, which meets the criteria of green chemistry. The results of a set of physicochemical methods provided information about an interesting structural-ordering property of calix[4]resorcinol in solution of larger open-chain molecules, which leads to the formation of joint nanoparticles. The obtained nanoparticles can increase the water solubility of hydrophobic antioxidants (quercetin and oleic acid), which allows increasing their biocompatibility. With an excess of RNA relative to calix[4]resorcinol, nanoparticles are formed that are capable of binding not only lipophilic substances, but hydrophilic doxorubicin. The obtained drug-bound forms exhibit enhanced cytotoxicity and penetrating ability into tumor cells. Despite the fact that these forms also demonstrated cytotoxic activity against normal cells, it can be said that the use of RNA and an oppositely charged macrocycle leads to joint aggregation, initiating the formation of promising nanoparticles for delivering both a chemotherapeutic agent and an antioxidant. The use of such nanoparticles in anticancer therapy will allow for the combined therapy of tumors by suppressing the development of tumors with doxorubicin and leveling the side effects with antioxidant.

## Figures and Tables

**Figure 1 ijms-24-07911-f001:**
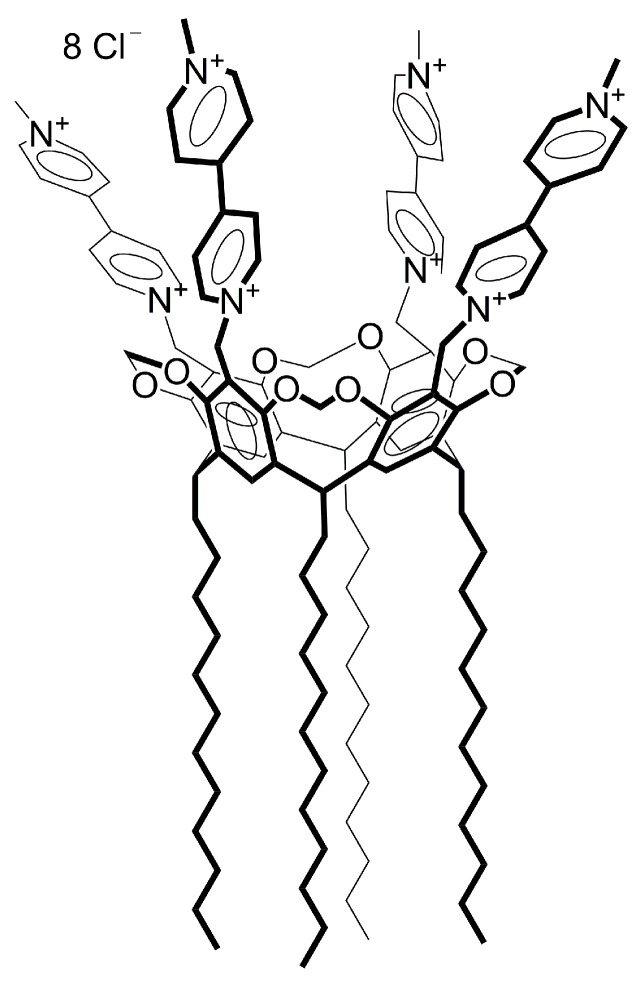
Chemical structure of VR.

**Figure 2 ijms-24-07911-f002:**
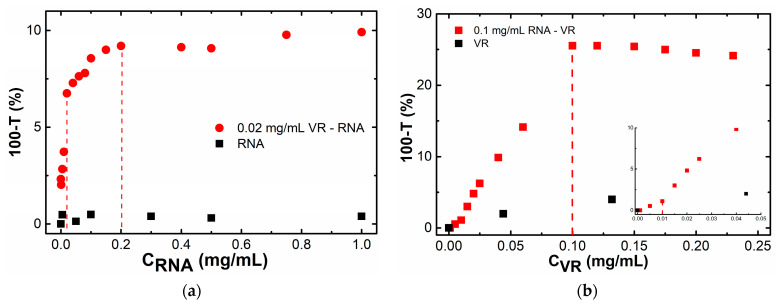
Dependence of the turbidity of solutions on the concentration of one component at a fixed content of the second: (**a**) C (VR) = 0.02 mg/mL, C (RNA) = 0–1 mg/mL; (**b**) C (RNA) = 0.1 mg/mL, C (VR) = 0–0.25 mg/mL.

**Figure 3 ijms-24-07911-f003:**
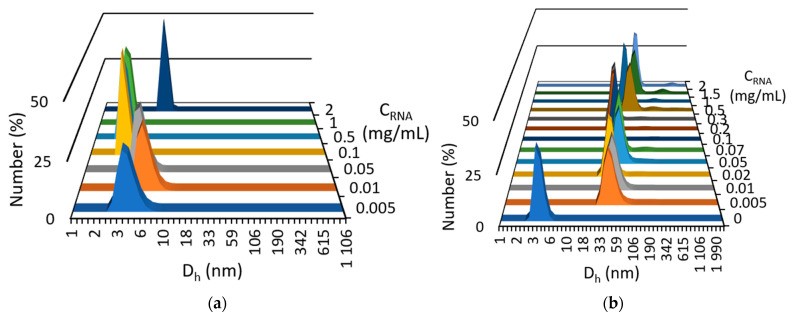
Particle size distribution (averaged over the number of particles) in solutions of pure RNA of various concentrations (**a**) and in the presence of 0.02 mg/mL VR (**b**).

**Figure 4 ijms-24-07911-f004:**
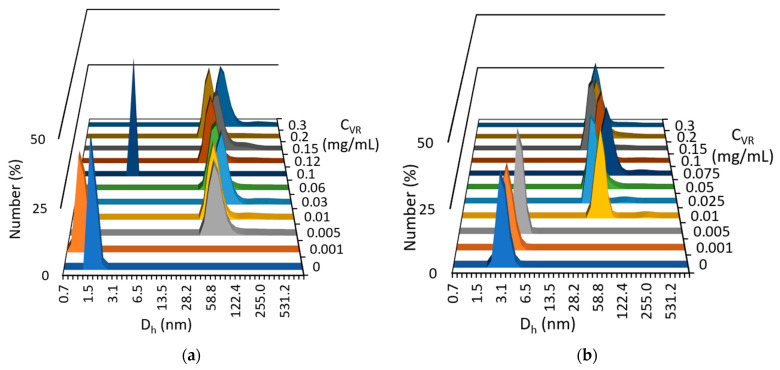
Particle size distribution (averaged over the number of particles) in solutions containing (**a**) 0.1 and (**b**) 0.5 mg/mL RNA and different amounts of VR.

**Figure 5 ijms-24-07911-f005:**
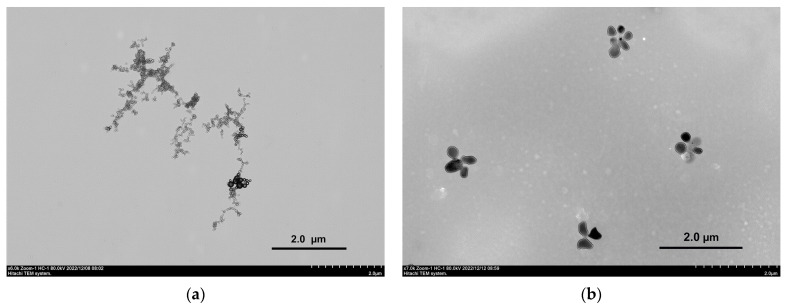
Transmission electron micrographs of RNA (**a**) and RNA:VR = 1:1 (**b**).

**Figure 6 ijms-24-07911-f006:**
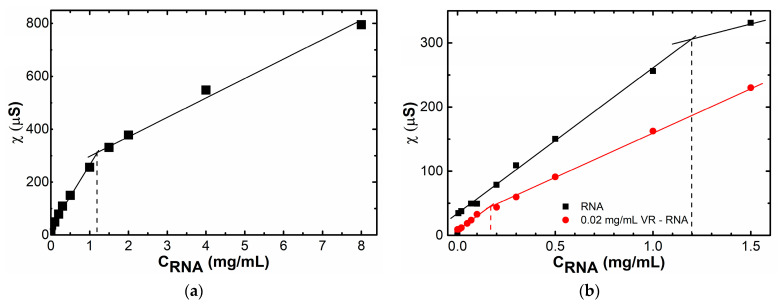
Specific conductivity of RNA (**a**) and VR–RNA (**b**) solutions, C (VR) = 0.02 mg/mL, C (RNA) = 0.005–8 mg/mL.

**Figure 7 ijms-24-07911-f007:**
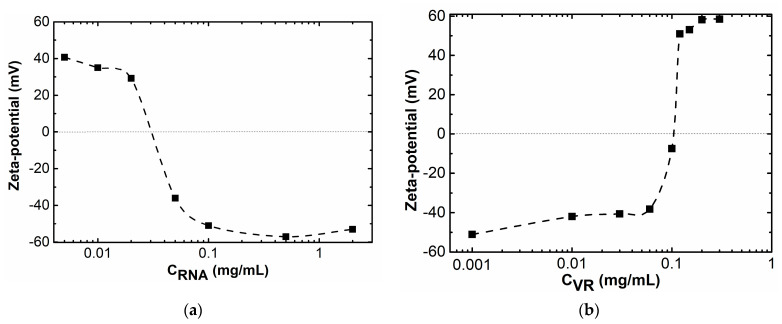
Dependence of the zeta potential of aggregates on the concentration of one component at a fixed content of the second: (**a**) C (VR) = 0.02 mg/mL, C (RNA) = 0.005–2 mg/mL; (**b**) C (RNA) = 0.1 mg/mL, C (VR) = 0.001–0.3 mg/mL.

**Figure 8 ijms-24-07911-f008:**
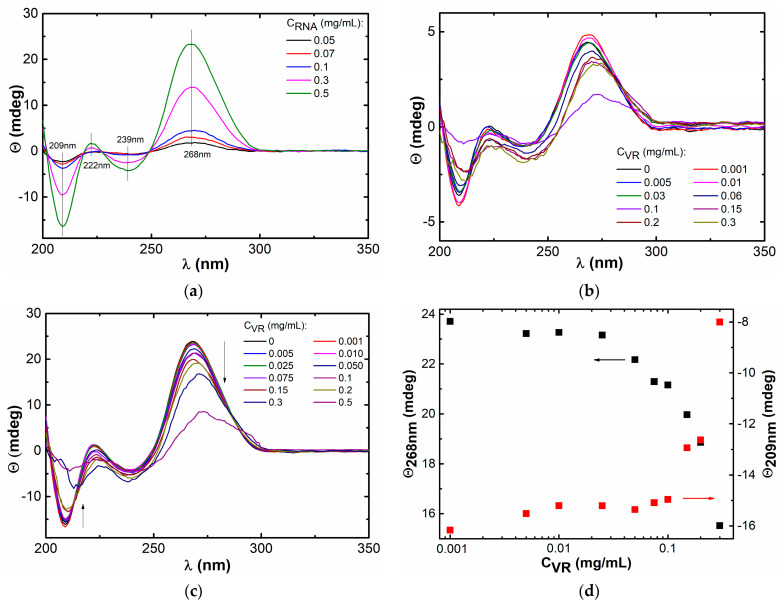
(**a**) CD spectra of pure RNA of various concentrations; (**b**) CD spectra of 0.1 mg/mL RNA in the presence of various amounts of VR; (**c**) CD spectra of 0.5 mg/mL RNA in the presence of various amounts of VR; (**d**) dependence of ellipticity at 268 and 209 nm on VR concentration in mixed solutions with 0.5 mg/mL RNA.

**Figure 9 ijms-24-07911-f009:**
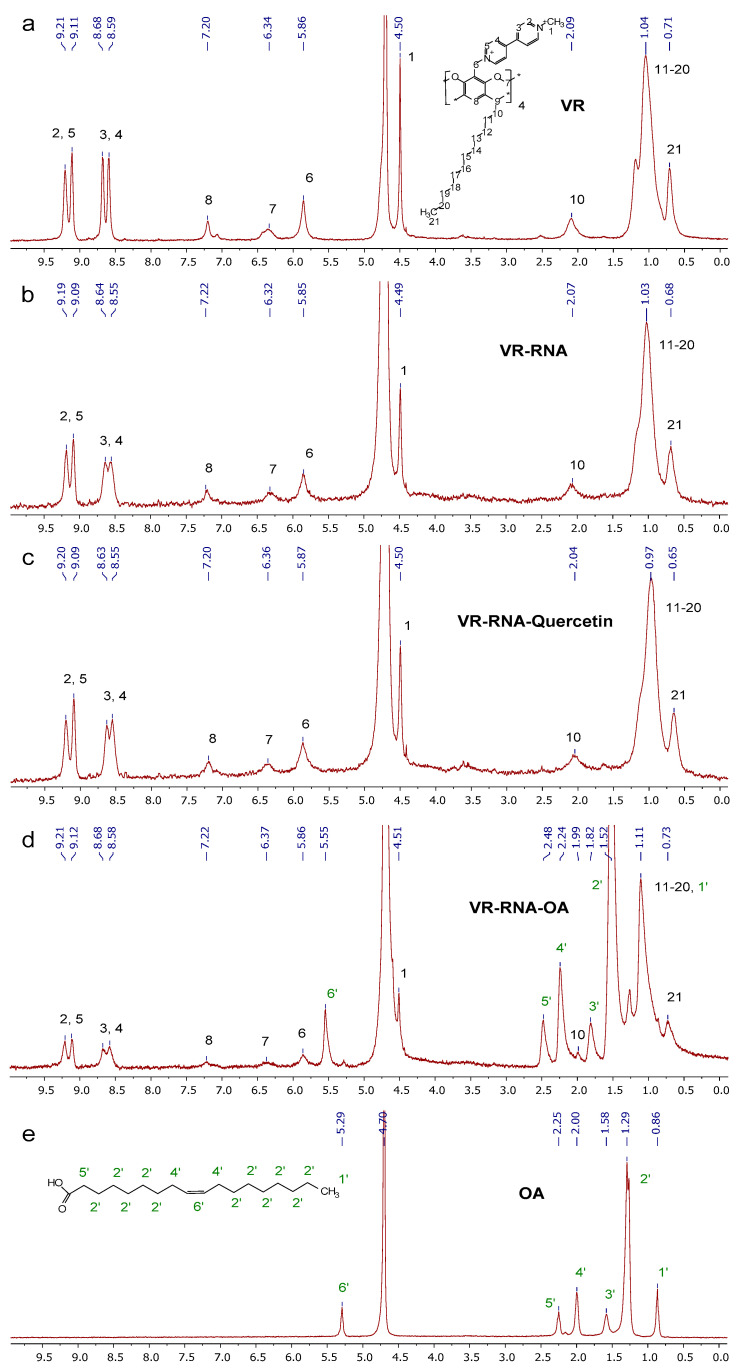
^1^H NMR spectra of 3 mg/mL VR (**a**), 3 mg/mL VR–2 mg/mL RNA complex in the absence (**b**) and in the presence of quercetin (**c**), oleic acid (**d**), and pure oleic acid (**e**) in D_2_O.

**Figure 10 ijms-24-07911-f010:**
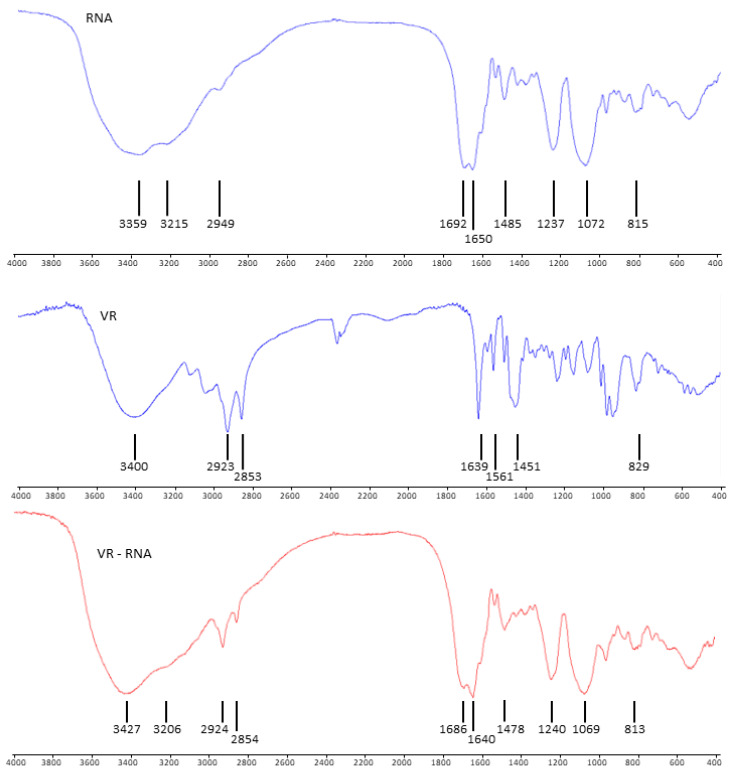
FTIR spectra of RNA, VR and VR-RNA complex.

**Figure 11 ijms-24-07911-f011:**
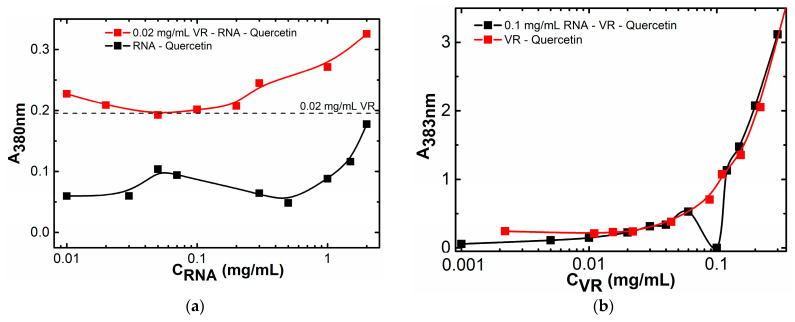
Dependence of quercetin absorption at 380 nm in aqueous solutions of (**a**) pure RNA and mixtures of RNA–VR with a constant VR concentration of 0.022 mg/mL on the concentration of RNA; (**b**) pure VR and mixtures of RNA–VR with a constant RNA concentration of 0.1 mg/mL on the VR concentration; optical path l = 1 cm.

**Figure 12 ijms-24-07911-f012:**
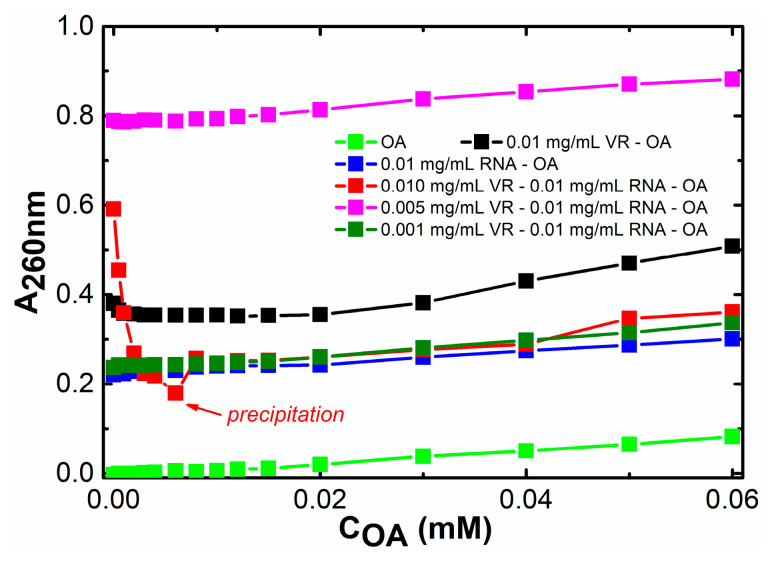
Dependence of absorbance at 260 nm in aqueous solutions of OA in the absence and presence of pure 0.01 mg/mL VR, pure 0.01 mg/mL RNA, mixtures 0.001 mg/mL VR–0.01 mg/mL RNA, 0.005 mg/mL VR–0.01 mg/mL RNA, 0.01 mg/mL VR–0.01 mg/mL RNA on OA concentration; optical path 1 cm.

**Figure 13 ijms-24-07911-f013:**
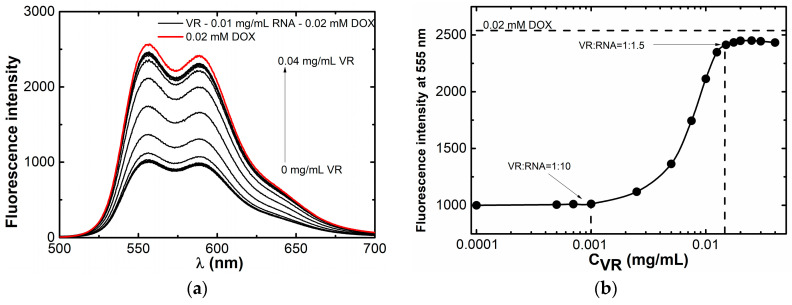
(**a**) Fluorescence spectra of 0.02 DOX in the presence of mixed solutions with a constant RNA concentration of 0.01 mg/mL and various amounts of VR, C(VR) = 0–0.04 mg/mL; (**b**) dependence of DOX fluorescence intensity at 555 nm in mixed solutions with a constant RNA concentration of 0.01 mg/mL on VR concentration.

**Figure 14 ijms-24-07911-f014:**
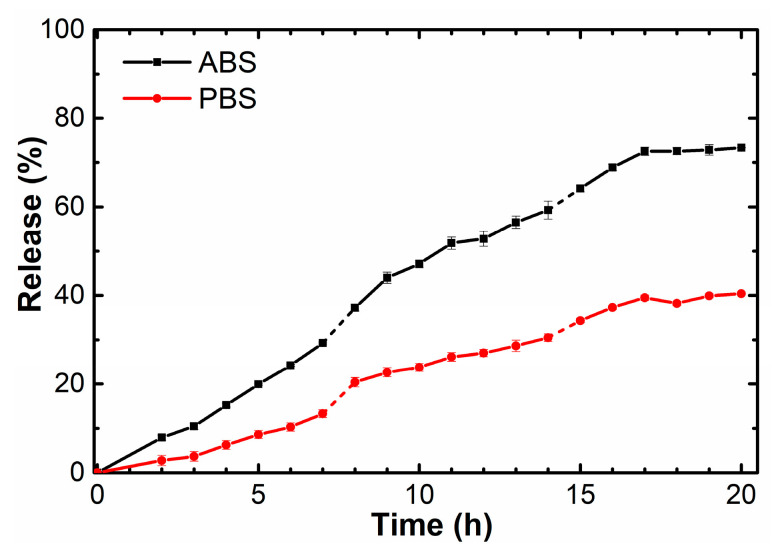
DOX release profiles from 0.02 mg/mL VR–0.2 mg/mL RNA aggregates in phosphate and acetate buffers.

**Figure 15 ijms-24-07911-f015:**
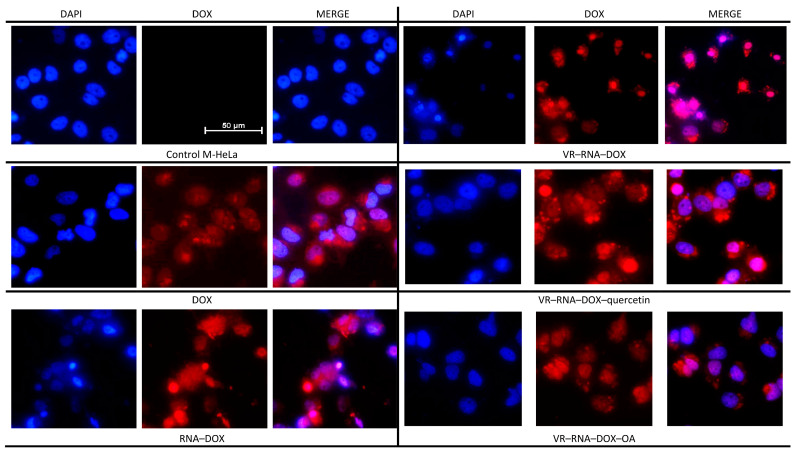
Analysis of DOX distribution in M HeLa cells treated with free DOX, DOX-loaded RNA, DOX-loaded VR–RNA, DOX-loaded VR–RNA–quercetin and DOX-loaded VR–RNA–oleic acid. Scale bar: 50 μm.

**Table 1 ijms-24-07911-t001:** IC_50_ values of pure RNA, RNA–VR mixtures in the absence and presence of quercetin and oleic acid, pure DOX, and DOX bounded in VR–RNA nanoparticles in the absence and presence of quercetin and oleic acid for normal (Chang liver) and tumor (M HeLa) cells.

Compound	M HeLa		Chang Liver	
	IC_50_ (RNA), mg/mL	IC_50_ (DOX), µM	IC_50_ (RNA), mg/mL	IC_50_ (DOX), µM
RNA	>0.1	-	>0.1	-
VR:RNA = 1:1	>0.01	-	>0.01	-
VR:RNA = 1:10	>0.1	-	>0.1	-
VR:RNA = 1:10	>0.1	-	>0.1	-
+ quercetin				
VR:RNA = 1:10 + OA	>0.1	-	>0.1	-
DOX	-	4.7	-	>10
RNA + DOX	0.09	9.0	>0.1	>10
VR:RNA = 1:10 + DOX	0.045	4.5	0.02	2.0
VR:RNA = 1:10	0.068	6.8	0.023	2.3
+ quercetin + DOX				
VR:RNA = 1:10	0.039	3.9	0.024	2.4
+ OA + DOX				

## Data Availability

Not applicable.

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
