# Peer review of "Emergence of Nanoscale Drug Carriers through Supramolecular Self-Assembly of RNA with Calixarene"

_ijms, 2023, doi:10.3390/ijms24097911_

Round 1
Reviewer 1 Report
Kashapov et al demonstrated a simple method for the creation of nanosized particles through the supramolecular self-assembly of RNA with a three-dimensional macrocycle from the calixarene family for the first time. This self-assembly into nanoparticles was realized using cooperative supramolecular interactions under mild conditions. The obtained nanoparticles were able to encapsulate various hydrophobic (quercetin, oleic acid) and hydrophilic (doxorubicin) drugs, as a result of which their cytotoxic properties were enhanced. This work demonstrated that intermolecular interactions between flexible RNA and rigid calixarene were promising for the bottom-up assembly of new soft-matter phases, expanding the design possibilities of nanoscale drug containers. Although it sounds interesting, some points need to be addressed.
I suggest adding some more characterizations, such as TEM or SEM.
I would suggest explaining the interactions with FTIR. By the way, what is the scenario of vaibility of cells treated with the DOx formulations. please explain.
Author Response
Kashapov et al demonstrated a simple method for the creation of nanosized particles through the supramolecular self-assembly of RNA with a three-dimensional macrocycle from the calixarene family for the first time. This self-assembly into nanoparticles was realized using cooperative supramolecular interactions under mild conditions. The obtained nanoparticles were able to encapsulate various hydrophobic (quercetin, oleic acid) and hydrophilic (doxorubicin) drugs, as a result of which their cytotoxic properties were enhanced. This work demonstrated that intermolecular interactions between flexible RNA and rigid calixarene were promising for the bottom-up assembly of new soft-matter phases, expanding the design possibilities of nanoscale drug containers. Although it sounds interesting, some points need to be addressed.
Comment: I suggest adding some more characterizations, such as TEM or SEM.
Response: The ordering of the RNA structure in the presence of VR was confirmed by TEM. The TEM image for pure RNA (Figure 5a) probably shows filamentous fragments of the biopolymer [10.1016/j.bpc.2009.05.004]. These filaments may be non-polar and therefore generally aggregate in an aqueous medium, forming bundles that can be attracted to each other due to the hydrophobic effect [10.1093/nar/gkl008]. The presence of an equimass amount of VR in an aqueous medium of RNA leads to the formation of more monodisperse spherical aggregates of a larger size (Figure 5b). The increased sizes of mixed aggregates obtained using TEM compared with DLS are due to the drying of the sample, during which the concentration increases. Nevertheless, comparison of images of pure RNA and RNA–VR mixture confirms the formation of joint nanoparticles due to the structural ordering action of the macrocycle. This point has been added in the revised manuscript.
Comment: I would suggest explaining the interactions with FTIR.
Response: To clearly prove the intermolecular nature of the complex formation between RNA and calix[4]resorcinol, we isolated the solid complex and examined it using IR spectroscopy (Figure 9). When comparing the IR spectrum of the complex with pure RNA and pure calix[4]resorcinol, band shifts are observed corresponding to complex formation between RNA and VR. The resulting IR spectra and their discussion have been added to the revised manuscript.
Comment: By the way, what is the scenario of vaibility of cells treated with the DOx formulations. please explain.
Response: The inhibition of cell viability is due to the mechanism of action of DOX manifested itself in the ability to bind and inhibit the enzyme topoisomerase II, which disrupts DNA repair and leads to apoptosis. Fluorescence microscopy experiments allowed us to evaluate the development of apoptotic effects in M HeLa cells induced by DOX both in free and encapsulated forms, and DOX encapsulation contributed to the intensification of apoptotic processes in these cells. This point has been added in the revised manuscript.
Reviewer 2 Report
The manuscript submitted by Kashapov et al. reports on a calix[4]resorcinol macrocyclic compound and its interactions with RNAs. The aim of the study is to demonstrate the capability of calix[4]resorcinol -RNA complex to encapsulate biologically active substances that has the potential of acting as dual drug delivery vector for anti-cancer treatment.
The manuscript starts with the presentation of the results obtained from turbidimetric measurements, which has been used to support results of DLS measurements. It has been thoroughly discussed that turbidimetric and DLS measurements are in close correlation and the formation of RNA-calix[4]resorcinol were associated with the appearance of secondary-type of particles in 400-700 nm range. While these results could be regarded as indirect proof of the molecular associations, there is no clear evidence of molecular nature of the complexation between RNA and calix[4]resorcinol. Similarly, encapsulation of the biologically active substances (namely quercetin, oleic acid and doxorubicin) by calix[4]resorcinol have been suggested based on spectrophotometric and zeta-potential measurements, again, there is no clear evidence of the “encapsulation” of the biologically active substances by calix[4]resorcinol.
While all the experimental results points out some sort of molecular interaction between calix[4]resorcinol, RNA and biologically active substances; I cannot agree on the potential role of the calix[4]resorcinol as a “nanoscale drug container” for anticancer treatment since there is no clear evidence on the role of calix[4]resorcinol to act as a drug-delivery vector.
The work seems to be based on the previously published results of the Authors (namely citation 45 and 46 in the manuscript). Thus, the results presented in the manuscript could be regarded as the continuation of the previous research on calix[4]resorcinol macrocyclic compound. Publication of the results in this manuscript may provide additional and valuable insights on the properties of calix[4]resorcinol. However, this manuscript requires additional experimental data to support the above criticized claims before it can be accepted for a publication.
Author Response
The manuscript submitted by Kashapov et al. reports on a calix[4]resorcinol macrocyclic compound and its interactions with RNAs. The aim of the study is to demonstrate the capability of calix[4]resorcinol -RNA complex to encapsulate biologically active substances that has the potential of acting as dual drug delivery vector for anti-cancer treatment.
Comment: The manuscript starts with the presentation of the results obtained from turbidimetric measurements, which has been used to support results of DLS measurements. It has been thoroughly discussed that turbidimetric and DLS measurements are in close correlation and the formation of RNA-calix[4]resorcinol were associated with the appearance of secondary-type of particles in 400-700 nm range. While these results could be regarded as indirect proof of the molecular associations, there is no clear evidence of molecular nature of the complexation between RNA and calix[4]resorcinol. Similarly, encapsulation of the biologically active substances (namely quercetin, oleic acid and doxorubicin) by calix[4]resorcinol have been suggested based on spectrophotometric and zeta-potential measurements, again, there is no clear evidence of the “encapsulation” of the biologically active substances by calix[4]resorcinol.
Response: Sorry for confusing you. The formation of RNA-calix[4]resorcinol were associated with formation of aggregates 50–70 nm in size in mixed solutions even with a low biopolymer concentration, below 0.5 mg/mL (Figure 3b). At an RNA concentration above 0.5 mg/mL, all viologen groups of the macrocycle interact with negatively charged RNA groups, resulting in the formation of nonionic aggregates capable of sticking together, which leads to the formation of particles with a large size of 400–700 nm. Additional explanatory edits have been made to the revised manuscript.
To clearly prove the intermolecular nature of the complex formation between RNA and calix[4]resorcinol, we isolated the solid complex and examined it using IR spectroscopy (Figure 9). When comparing the IR spectrum of the complex with pure RNA and pure calix[4]resorcinol, band shifts are observed corresponding to complex formation between RNA and VR. The resulting IR spectra and their discussion have been added to the revised manuscript.
We consider the presence of the characteristic absorption band of quercetin in the UV spectra, which are added as a separate Figure S1 in the supplementary material, as fairly clear evidence of the encapsulation of this flavonoid. Quercetin in pure water does not give any absorption on the UV spectrum, and its solubilization observed spectrophotometrically in compositions formed in an aqueous medium is a confirmation of the presence of a hydrophobic domain in these compositions [10.1016/j.foodchem.2016.05.034, 10.1016/j.foodhyd.2020.106449]. Taking into account the hydrophobic nature of quercetin, the appearance of its absorption band in the aqueous medium of the calixarene-RNA mixture convincingly indicates the encapsulation of quercetin in the hydrophobic domains of the aggregates formed in this mixture. Yes, unfortunately, spectrophotometric methods do not allow for similar studies of direct confirmation of the encapsulation of oleic acid due to the absence of chromophore and auxochromic groups in the structure of this acid. However, a sufficiently significant change in the intensity of the absorption band of calixarene under the action of oleic acid can be reliable evidence of the encapsulation of this acid based on comparison with a wide range of other compositions for which the effect of the addition of oleic acid has also been studied. Unfortunately, there is no similar study of the binding of oleic acid by calixarene in the literature, so we cannot cite the literature supporting this point. UV spectra for formulations with oleic acid are also added to the supplementary manuscript in Figure S3.
When studying the binding of doxorubicin, its interaction with pure RNA was studied using fluorescence, which is consistent with a large amount of literature data. One example of detecting the binding and release of doxorubicin in systems using fluorescence is [10.2147/IJN.S135086]. Therefore, based on these literature data, we can confidently state the observed processes of doxorubicin binding to calixarene-RNA aggregates and release.
Comment: While all the experimental results points out some sort of molecular interaction between calix[4]resorcinol, RNA and biologically active substances; I cannot agree on the potential role of the calix[4]resorcinol as a “nanoscale drug container” for anticancer treatment since there is no clear evidence on the role of calix[4]resorcinol to act as a drug-delivery vector.
Response: Yes, we fully agree with you that a potential role as a nanoscale drug container is played not by calix[4]resorcinol, but by a supramolecular system based on RNA and calix[4]resorcinol. We have supplemented the revised manuscript with TEM images (Figure 5), in the discussion of which the structural ordering role of the macrocycle is noted, leading to the formation of joint aggregates at a certain ratio of RNA:calix[4]resorcinol. A minor clarification about this has been made to the abstract of the revised manuscript.
Regarding the vector properties, we should note such an interesting and no less important property of viologen molecules, namely paraquat, as targeting mitochondria [Zhao G., Cao K., Xu C. et al. Int. J Biol. sci. 2017, 13 888-900]. The metabolism of cancer cells often shifts from oxidative phosphorylation to aerobic glycolysis as the main generator of cellular ATP, which leads to an increase in the amount of NADH that can be oxidized by viologen groups [Yu G., Yu W., Shao L., Zhang Z., Chi X., Mao Z., Gao C., Huang F. Adv. Funct. mater. 2016, 26, 8999–9008]. The reduction of a dicationic viologen to a cationic radical and neutral forms may be accompanied by the release of a drug encapsulated in aggregates that were formed due to electrostatic interactions of viologen groups of macrocycle with oppositely charged molecule. This point has been added in the Introduction section of the revised manuscript.
Comment: The work seems to be based on the previously published results of the Authors (namely citation 45 and 46 in the manuscript). Thus, the results presented in the manuscript could be regarded as the continuation of the previous research on calix[4]resorcinol macrocyclic compound. Publication of the results in this manuscript may provide additional and valuable insights on the properties of calix[4]resorcinol. However, this manuscript requires additional experimental data to support the above criticized claims before it can be accepted for a publication.
Response: Paper #45 refers to the study of the self-assembly of viologen calixarenes, which differ in the length of the alkyl fragments on the lower rim, individually in water, and paper #46 includes a study of the interaction of these calixarenes with SDS. Yes, the work presented is based on these works, because thanks to paper #45 we know the critical aggregation concentration of macrocycle, and thanks to paper #46 we know about possible supramolecular interactions between oppositely charged macrocyclic and open chain amphiphiles. The publication of the results in the present manuscript provided additional and valuable information about the interesting structure-ordering property of calix[4]resorcinol with respect to larger open-chain molecules, leading to the formation of joint nanoparticles. This point has been added in the Conclusions section of the revised manuscript.
Round 2
Reviewer 2 Report
I would like to thank Authors for the additional experimental data and the revision of the manuscript. However, I still have difficulties to understand the hypotheses of the Authors. TEM imaging and FTIR spectroscopy can provide valuable results to support the hypothesis, however, such methods still require additional support to clarify the molecular phenomenon that Authors suggest. Unfortunately, only two images from TEM cannot be regarded as full-proof for RNA aggregation/complexation, since the sample preparation for TEM, itself, can always lead formation of complex structures (although similar molecular complexations are not possible in solution). Moreover, FTIR spectra of the samples actually do not provide any direct proof of the molecular associations since the spectrum of the pure RNA consist of broadened, strong bands which overlaps with the bands observed for the RNA-calix[4]resorcinol complex. Without deconvolution of spectra by mathematical means, I cannot agree with the claims made over FTIR spectra.
I would like underline that the major problem of the manuscript is the bold claims made over "molecular drug containers" and "encapsulation" of the biologically active substances. I would like to repeat that, even after the revision, the experimental data do not support such hypotheses. Without NMR, mass spectroscopy, gel electrophoresis of the complexes followed by both qualitative and quantitative spectroscopy of the extracts or (in the best case scenario) single-crystal X-ray diffraction data of the complexes, the current form of the manuscript is just full of speculations. I can believe any sort of interaction between RNA, VR and biologically active substances, however, currently available data is far from convincing the readers for existence of any sort of "encapsulation" mechanism or VR-RNA complexation in solution state.
Author Response
Comment: I would like to thank Authors for the additional experimental data and the revision of the manuscript. However, I still have difficulties to understand the hypotheses of the Authors. TEM imaging and FTIR spectroscopy can provide valuable results to support the hypothesis, however, such methods still require additional support to clarify the molecular phenomenon that Authors suggest. Unfortunately, only two images from TEM cannot be regarded as full-proof for RNA aggregation/complexation, since the sample preparation for TEM, itself, can always lead formation of complex structures (although similar molecular complexations are not possible in solution). Moreover, FTIR spectra of the samples actually do not provide any direct proof of the molecular associations since the spectrum of the pure RNA consist of broadened, strong bands which overlaps with the bands observed for the RNA-calix[4]resorcinol complex. Without deconvolution of spectra by mathematical means, I cannot agree with the claims made over FTIR spectra.
Response: The intermolecular interaction between RNA and calix[4]resorcinol was confirmed by a set of physicochemical methods. The results obtained by these different methods are in good agreement with each other. Our manuscript has been supplemented with valuable 1H NMR spectroscopy data to confirm intermolecular interactions between VR and RNA in the water. Since the equimass complex VR:RNA = 1:1 precipitated in aqueous solution due to the high concentrations required to obtain 1H NMR spectra, a mixture with a slight 1.5-fold excess of VR was examined using 1H NMR (Figure 9). When comparing the 1H NMR spectrum of this mixture (Figure 9b) with that of pure VR (Figure 9a), a decrease in signal intensity is observed as a result of co-aggregation. The largest chemical shift by 0.04 ppm is experienced by viologen protons distant from positively charged nitrogen atoms, which may be due to a change in the environment of viologen ions as a result of electrostatic interaction with RNA. Upfield shift of protons of the VR terminal methyl group by 0.03 ppm and proton broadening of the VR dodecyl chain in the presence of RNA indicate the participation of macrocycle alkyl chains in interaction with RNA, leading to joint aggregation. This point has been added in the revised manuscript.
Comment: I would like underline that the major problem of the manuscript is the bold claims made over "molecular drug containers" and "encapsulation" of the biologically active substances. I would like to repeat that, even after the revision, the experimental data do not support such hypotheses. Without NMR, mass spectroscopy, gel electrophoresis of the complexes followed by both qualitative and quantitative spectroscopy of the extracts or (in the best case scenario) single-crystal X-ray diffraction data of the complexes, the current form of the manuscript is just full of speculations. I can believe any sort of interaction between RNA, VR and biologically active substances, however, currently available data is far from convincing the readers for existence of any sort of "encapsulation" mechanism or VR-RNA complexation in solution state.
Response: The revised manuscript has been supplemented with a point regarding the complete lack of absorption for pure quercetin in water. The black line in Figure S1 of the Supplementary Material indicates that quercetin in pure water does not give any absorption in the 380 nm region due to its lack of dissolution. Therefore, the presence of quercetin absorption at 380 nm in UV spectrum of aqueous solutions of amphiphiles cannot be a speculation, but is clear evidence of the solubilization of quercetin into the hydrophobic domains of the formed aggregates, which is confirmed by a considerable amount of literature data. Only some of these literature data regarding quercetin can be seen in 10.1016/j.molliq.2022.119596, 10.1016/j.molliq.2017.11.044, 10.1002/biot.201700389, etc. To further confirm the encapsulation of quercetin in aggregates based on VR and RNA, 1H NMR spectroscopy was used (Figure 9c). When comparing the 1H NMR spectra of the VR–RNA mixture in the absence (Figure 9b) and in the presence of quercetin (Figure 9c), the largest chemical shift by 0.06 ppm is observed for protons of alkyl chains of VR, underscoring the location of quercetin in the hydrophobic domains formed by these chains. Similar changes in chemical shifts are observed when comparing the 1H NMR spectra of oleic acid in the absence (Figure 9e) and in the presence of a mixture of VR-RNA (Figure 9d), which strongly indicates the encapsulation of oleic acid in nanoparticles based on VR-RNA. According to this comparison, the largest change in the chemical shifts of alkyl chain protons among other VR protons in the presence of OA and the greater change of all OA protons in the presence of VR–RNA mixture strongly confirm the encapsulation of OA in aggregates based on VR–RNA due to solubilization into hydrophobic domains.
Binding of DOX was reiterated by comparing its 1H NMR spectrum (Figure S5a) with that of its mixture with RNA (Figure S5b). The intensity of most DOX proton signals in this mixture was reduced due to electrostatic interaction with RNA, and some DOX proton signals were broadened and overlapped by broad RNA proton signals. The only clearly observed high-field shift of the methyl group near the positively charged amino group of DOX indicates its binding to RNA. The presence of VR in a tenfold deficiency relative to RNA in the composition of DOX did not affect the value of this chemical shift, which confirms the bound state of the drug in aggregates based on VR–RNA and correlates with fluorometry data (Figure S5c). Unfortunately, increasing the proportion of VR in this composition led to precipitation due to the high concentrations of the components required to obtain 1H NMR spectra. All these points regarding the binding of biologically active substrates have been added to the revised manuscript.
Mass spectrometry (MALDI and ESI) was used by us, but, unfortunately, it did not help to give valuable results due to the large number of peaks caused by the polymeric nature of RNA. We agree with you that gel electrophoresis should be performed due to its wide application in the study of nucleic acids, but again, unfortunately, our institute is unable to conduct this experiment due to lack of equipment. We hope that data obtained by a set of physicochemical methods available to us might suit you.
Round 3
Reviewer 2 Report
I would like to thank Authors for providing additional experimental data. Certainly, 1H-NMR data provides additional insides on the Authors' hypotheses about the molecular interactions between VR, RNA and biologically active substances.
It is now even clearer that the some sort of molecular interaction exists between the molecules, yet, the provided that still do not indicate the existence of "molecular encapsulation" or a molecular assembly that act as "drug container". If the Authors would like to keep such claims, they should consider to outsource mass spectroscopy and electrophoresis experiments and include them to the manuscript as full proof of their hypotheses.
At this point, without the mass spectroscopy data, I can only recommend Authors to reconsider their claims and admit that there are not enough experimental evidence to support their claims on encapsulation or molecular drug inclusion. I would recommend the title of the manuscript to be changed, the word "encapsulation" to be entirely removed from the manuscript and to revise the manuscript only on the hypotheses of "strong supramolecular interactions promising for biological applications".
Author Response
When studying large oligonucleotides using mass spectrometry, there are problems of increased complexity of their ionization and efficient fragmentation [doi.org/10.1016/j.tibs.2006.01.004]. The polymeric nature of the yeast RNA we used makes it difficult to interpret the mass spectra due to the large set of observed peaks. A reliable mass spectrum of the complex can be obtained by using one macrocycle to bind one water-soluble molecule, which was shown by us using ESI in [doi.org/10.1134/S1023193513040150]. Unfortunately, a similar peak of the complex with water-soluble doxorubicin could not be obtained in the presented manuscript due to the complication of the supramolecular system by the biopolymer. We can agree with you that the use of the terms "molecular encapsulation" and "drug container" is not entirely correct for the case of hydrophilic doxorubicin, since the aggregates are formed in water, and the location of doxorubicin in the aggregates must certainly be on their surface facing towards water phase. However, in the case of using hydrophobic substrates, which should be located in the hydrophobic domains of the formed aggregates, one can assume their unambiguous binding within these aggregates. Since there are no mass spectrometry data in the manuscript, we can only exclude such terms as "container" and "encapsulation" from the manuscript. These terms have been replaced in the revised manuscript, and all edits are highlighted in turquoise.
Round 4
Reviewer 2 Report
I do understand the difficulties in obtaining mass spectroscopy data. However, speaking of the complex supramolecular structures as reported in the manuscript, we cannot think of any other method, which is more accessible and starightforward, than mass spectroscopy.
I think that manuscript could be accepted after the final revision of the Authors, where the terms "molecular encapsulation" and "drug container" were replaced by appropriate terms that are supported by the currently available experimental data.
I wish to see the continuation of this research work in the future, hopefully supported by mass spectroscopy data (or even with more daring single-crystal XRD approach. Let me note that it is not impossible).